# Executive Functions and Attention Processes in Adolescents and Young Adults with Intellectual Disability

**DOI:** 10.3390/brainsci11010042

**Published:** 2021-01-03

**Authors:** Tommasa Zagaria, Gabriella Antonucci, Serafino Buono, Marilena Recupero, Pierluigi Zoccolotti

**Affiliations:** 1Oasi Research Institute-IRCCS, 94018 Troina, Italy; fbuono@oasi.en.it (S.B.); mrecupero@oasi.en.it (M.R.); 2Department of Psychology, Sapienza University of Rome, 00185 Rome, Italy; gabriella.antonucci@uniroma1.it (G.A.); pierluigi.zoccolotti@uniroma1.it (P.Z.); 3IRCCS Santa Lucia Foundation, 00179 Rome, Italy

**Keywords:** executive functions, intellectual disability, attentional processing, speed of processing

## Abstract

(1) Background: We made a comprehensive evaluation of executive functions (EFs) and attention processes in a group of adolescents and young adults with mild intellectual disability (ID). (2) Methods: 27 adolescents and young adults (14 females and 13 males) with ID, aged between 15.1 and 23 years (M = 17.4; SD = 2.04), were compared to a control group free of cognitive problems and individually matched for gender and age. (3) Results: As for EFs, individuals with ID were severely impaired on all subtests of the Behavioral Assessment of Dysexecutive Syndrome (BADS) battery. However, we also found appreciable individual differences, with eight individuals (approximately 30%) scoring within normal limits. On the attention tests, individuals with ID were not generally slower but presented specific deficits only on some attention tests (i.e., Choice Reaction Times, Color Naming and Color–Word Interference, and Shifting of Attention for Verbal and for Visual Targets).The role of a global factor (i.e., cognitive speed) was modest in contributing to the group differences; i.e., when present, group differences were selectively associated with specific task manipulations, not global differences in cognitive speed. (4) Conclusions: The study confirmed large group differences in EFs; deficits in attentional processing were more specific and occurred primarily in tasks taxing the selective dimension of attention, with performance on intensive tasks almost entirely spared.

## 1. Introduction

The executive functions (EFs) domain refers to a set of cognitive mechanisms that allow optimizing performance and identifying the most appropriate behaviors aimed at achieving a specific purpose through the selection and simultaneous activation of different processes [1]. This domain includes skills such as planning, inhibitory control, decision-making processes that support the selection of functional response and its modification in relation to changing environmental contingencies, self-monitoring, attentional control, and working memory (e.g., References [2,3]). One dominant view is that three moderately interrelated functions account for most individual variability in EF: mental set shifting, information updating and monitoring, and inhibition of prepotent responses [4]. Therefore, the executive functions control and modulate basic cognitive functions and other functions and behaviors that are fundamental for adaptation to the environment [1].

A related area concerns the evaluation of attention processes. In general, attention processes refer to the quality of the information processing active at any given time. Both selectivity and intensity dimensions are critical to account for a given performance. Spikman and van Zomeren [5] described attention as “the state of a processing system that is optimally tuned in terms of selectivity and intensity”. The selectivity dimension refers to the process of giving priority to specific stimuli with respect to others; it can be tested with tasks, such as Focused or Divided Attention. The intensity dimension refers to the process of activating and maintaining attention over time. Related tasks refer to phasic or tonic changes in alertness as well as to the loss of performance over extended periods of time (i.e., sustained attention tasks) [5]. A related concept concerns the speed of information processing. This is important because too much information is usually available for ongoing processing and the observer needs to select relevant information. Note that EF and attention are typically considered as contiguous but partially distinct areas requiring separate (but sometimes partially overlapping) measures. Notably, recent research tends to link EF and attention both developmentally and in terms of brain activation [6]. For example, it has been found that the development of early attentional processes provides a foundation for later EF through the larger mechanism of self-regulation [7,8]. By contrast, most research in clinical populations (i.e., intellectual disability (ID), which is the focus of the present study) has considered either EF or attention. In fact, very few studies have examined both areas of functioning in the same population (see below).

Deficiencies in EF, as well as in attention processes, have very negative effects on adaptation, such as impaired self-control, emotional instability or difficulties in planning, particularly in individuals with intellectual disabilities (for a review, see Reference [9]). Furthermore, attentional problems have important and widespread consequences on everyday activities. One area which has been the object of much research is that of return to work, which is frequently hindered in individuals with attentional disturbances (e.g., for a review, see Reference [10]). 

Research on executive functions, which are frequently referred to as “frontal functions” [11], together with memory and attention, i.e., the basic cognitive functions that also belong to the frontal lobes, are recognized as particularly important for learning in children and adolescents with mild ID [9]. Analyses aimed at investigating the correlation between intellectual quotient (IQ) and EF have found that some components of intellectual functioning correlate more with EF than others [12]. In studies of individuals with ID, a greater correlation was found between verbal IQ and EF [13]. Furthermore, it has been reported that similar IQs may correspond to different deficits in EF [14], while similar EF may correspond to different IQs [15].

The study of EFs in individuals with ID has been primarily carried out in persons with particular syndromic pictures, i.e., autism [16], fragile X syndrome (e.g., References [17,18,19,20]), Down syndrome [14,21,22,23,24,25], Prader–Willi syndrome [20,26,27,28] and Williams syndrome [25,29]. Deficits in planning capacity [18,23,24,29], inhibition mechanisms [18,23,24,29], and working memory [18,23,24,30] are common particularly in Down, fragile X, and Williams syndromes. Moreover, a few studies have compared differences among syndromes; for example, greater deficits in shifting and verbal aspects of memory and inhibition were reported in persons with Down syndrome while greater deficits in planning were detected in persons with Williams Syndrome (e.g., References [25]).

There is also a long tradition of studies relating intelligence and attention both in the normal range and comparing individuals with different degrees of intellectual disability with chronologically or mentally matched typically developing individuals. One early hypothesis was that speed of information processing was the basis of general intelligence (e.g., References [31,32,33]; or at least one of the two routes crucial to knowledge acquisition [34]. In this view, measures of speed of processing, such as reaction times (RT) or inspection times (i.e., the minimum time required to make a simple perceptual judgement; Reference [32]), have a significant relationship with IQ [35].

Various studies have also examined the performance in attention tasks of individuals with different degrees of intellectual disabilities [36]. It is difficult to evaluate this research due to the co-presence of various medical conditions, such as ADHD, autism or externalizing disorders. Consequently, it is not always easy to establish whether attentional deficits should be specifically linked to the presence of ID. Thus, while early research pointed to a close association between attention deficits and ID (for a review, see Reference [37]), Burack et al. [38] re-examined this literature and found it particularly deficient for various methodological reasons, and commented on the “mysterious myth of attention deficits and other defect stories” in the case of ID. Still, more recent research has confirmed the presence of attentional difficulties [39,40]. However, one complexity in examining this literature is to differentiate the effect of intellectual disability from other concurrent condition, such as the presence of ADHD [41] or conduct disorder [40]. 

Even in recent years, many studies have focused on one or a few measures of attention (e.g., References [40,42]). More recent approaches have emphasized that attention refers to a complex set of processes (for an extensive discussion, see Reference [43]) and that there is no such thing as a single measure of attention [5]. In particular, van Zomeren and Brouwer [43] referred to both tonic and phasic alertness and vigilance as critical measures of the intensive dimension and to selective and divided attention as markers of the selective dimension. This view was instrumental in guiding research on patients with both acquired brain lesions (e.g., Reference [44]) and neurodevelopmental disorders [45].

Speed of processing is certainly related to attentional tasks, but opinions on how to measure this construct have changed considerably over the years. As stated above, much early research searched for single measures, such as RTs to simple visual stimuli or inspection times, that could provide a reliable index of individual speed of processing (e.g., References [31,32,33]). A different perspective comes from studies that tried to disentangle global versus task-specific contributions to individual differences on timed tasks [46,47]. For example, according to Faust et al. [46] individual performance on timed tasks can be explained as in terms of the multiplicative interaction of two factors: the rate (i.e., the individual cognitive speed of the individual) and the amount of processing (i.e., the difficulty of a given task).Their rate and amount model (RAM) makes explicit predictions about how to determine these two components (which are present in all tasks).In particular, cognitive speed can be evaluated at both an individual and a group level as the proportional increase in performance as a function of the increase in task difficulty. In practice, this approach requires plotting the performance of a slow group against that of a control group by means of a Brinley plot (e.g., Reference [48]) and examining whether a single multiplicative factor accounts for a proportion of the group differences. In this perspective, cognitive speed can be seen as an underlying trait dimension that affects individual performance across all timed tasks (hence no single task can detect it) and should be viewed as conceptually different from attention processes which by definition are closely coupled with the type of ongoing processing required within a specific task condition. 

Early information about this aspect comes from a meta-analysis carried out by Kail [49], who examined the role of the general slowing of information processing by contrasting the performance of persons with ID versus control groups over a large set of experimental group data points in attention studies (overall considering a total of 518 pairs). Unlike RAM, Kail [49] focused only on the measure of beta values (i.e., the slope of the relationship linking the performance of the two sets of groups). He reported an average value of 1.43 for individuals between 12 and 16 years of age and 1.48 in the 39–45 age range; for individuals between 17 and 33 years of age values were appreciably higher (ranging from 1.65 to 2.34) but varied with IQ level. It should be observed that many of the studies taken into consideration by Kail [49] go back to the sixties and seventies; as we have seen above, early studies on attention processing have been severely criticized with regard to a number of methodological shortcomings [38]. Thus, conclusions based on Kail’s meta-analysis should be taken with caution (further comments on this point are presented in the discussion). Overall, examining cognitive speed may well complement analysis of the intensity and selectivity dimensions which van Zomeren and Brouwer [43] postulated as the underlying dimension of attention processes.

The present study is based on the observation that most research on EF and attention in intellectual disabilities was carried out in parallel and very few studies examined both sets of processes, even though they are closely interwoven. Furthermore, as to attention, several studies focused only on one or a few measures and failed to provide a comprehensive evaluation of both the selectivity and intensity dimensions of attention (e.g., References [40,42]. Finally, it is difficult to reach a conclusion about the contribution of individual cognitive speed to performance on timed tasks as the only available review is based on updated research [49]. However, recent models underscore the importance of accounting for both global and specific components in timed tasks. Notably, different models of individual performance (such as the RAM [46] or the Difference Engine Model (DEM) [47]) offer specific predictions to this aim.

In the present study, we set out to provide a comprehensive evaluation of EF and attention processes of adolescents and young adults with mild intellectual impairments. An effort was made to use a variety of EF and attention tasks that cover a spectrum of critical processes. Furthermore, in keeping with the observation by Burgess and colleagues [50,51] that laboratory EF measures may be insensitive and inappropriate for detecting the EF capacities of individuals with ID, a battery of ecologically valid tests was used (Behavioral Assessment of the Dysexecutive Syndrome (BADS); Reference [52]). In the case of attention, a variety of tasks (including several RT measures) was used so as to allow testing both the selectivity and intensity dimensions as well as the putative presence of global components in the group differences between individuals with ID and controls.

## 2. Materials and Methods

### 2.1. Sample

The study sample was recruited from individuals attending the IRCCS Oasi Maria SS in Troina (Italy). The sample was composed of 27 adolescents and young adults (14 females, 13 males) with ages ranging between 15.1 and 23 years (M = 17.4; SD = 2.04) and with a diagnosis of mild ID according to the criteria of the DSM-5. The participants did not present genetic syndromes; six individuals had an associated diagnosis of an anxiety disorder, two of a behavior disorder, one of an obsessive–compulsive disorder and three of epilepsy. Informed consent and privacy consent forms were signed by all participants prior to enrolment.

The experimental sample was compared to a control group, individually matched for gender and age and free of cognitive problems. The control group included 27 individuals (14 females and 13 males) whose ages ranged from 15 to 23 years (M = 17.57; DS = 2.06). The students were volunteers from local high schools and colleges. At the time of participation, they completed an anamnestic questionnaire in use in our center; none of them reported having been diagnosed with learning or other developmental disorders or having received cognitive treatment or special support in school.

### 2.2. Instruments

Intellectual functioning was evaluated using the Wechsler Intelligence Scale. Depending on the chronological age of the participants, the Wechsler Intelligence Scale for Children—Fourth Edition (WISC-IV) was administered to 12 adolescents and the Wechsler Adult Intelligence Scale IV (WAIS-IV) to 15 young adults. Similarly, the WISC-IV was given to 12 control adolescents and the WAIS-IV to 15 young adults. Both WISC-IV and WAIS-IV were administered following standard procedures; scoring was carried out using Italian standard norms [53,54]. 

We adopted the Behavioral Assessment of Dysexecutive Syndrome (BADS; ^©^1996 by Pearson) battery for the evaluation of executive functions [52] and the Attention and Concentration Battery (^©^2000 by Erickson, Italy) for the detection of attention processes [55].

#### 2.2.1. Behavioral Assessment of the Dysexecutive Syndrome (BADS)

The BADS [52] (Italian normalization [56]) is an ecological measure of the executive functions involved in everyday life related to cognitive flexibility, inhibitory control, planning, judging, monitoring, estimating and organizing time.

The six subtests that comprise the BADS refer to different cognitive abilities. The first subtest, the “Rule Shift Card” (RSC) reflects a “Cognitive flexibility” capacity. Participants are asked to learn a rule and apply it to a task. Then they are told to forget the previous rule, learn a new one and apply it to the old task. This is a simple measure of the ability to shift from one rule to another and to keep track of some features of the task and a certain rule. The score is the number of errors with a correction for the time taken.

The second is a problem-solving subtest, i.e., “Action Programme” (AP). Participants are asked to remove a small cork from a tube using any of the objects on the table but without touching the lid with their fingers. To solve the task 5 steps must be taken. The key is to use the water to make the cork float to the top of the tube. The score is the number of correct steps taken to solve the problem with a correction for the time taken.

In the “Key Search” (KS) subtest participants are presented with an A4-size sheet of paper with a square in the middle and a small black dot below it. They are told that the square is a large field in which they have lost their keys; they are asked to draw a line to show where they would walk to search the field and to make sure they would find the keys. The score on this task is based on the appropriateness and efficacy of the strategy adopted (ranging from 0 to 13) with a correction for the time taken.

The fourth subtest, i.e., “Temporal Judgement” (TJ) requires that the participants make a reasonable estimate of the time common people need to do four things. The four questions are as follows: (1) How long does a routine dental check-up take? (2) How long do most dogs live? (3) How long does it take to blow up a party balloon? (4) How long does it take to make an Italian coffee? This test explores cognitive estimation ability. The score on this task is the number of correct answers on the test.

The “Zoo Map Test” (ZMT) subtest is a navigational planning task. The participants are required to show how they would visit a zoo obeying a number of rules. There are two trials; in both, the participants have to plan a visit. The first trial consists of a high demand version of the task; the second trial is a low demand task and is easier than the first. The comparison of performances on the two trials allows evaluating the participants spontaneous planning ability when the structure is minimal (first trial) versus the ability to follow a concrete externally imposed strategy (second trial) when the structure is high. The score (sum of parts 1 and 2) is based on the difference between the correct sequence of the eight places to visit and the number of errors, the planning time on the second part and the total time on the second part.

The last subtest is called “Modified Six Elements” (MSE); it makes demands on the ability to plan, organize and monitor behavior and to remember to carry out an intention at a future time. The participants are required to perform three trials, each of which is divided into two parts (A and B), and to attempt to perform at least some of the six subtests in a ten-minute period. Only one rule must be observed, i.e., the two A and B parts of a given task cannot be carried out in sequence. This test explores the ability to organize behavior in predefined time intervals. The score is based on the number of tasks attempted minus the number of rule violations made (ranging from 0 to 6) with a correction for the time taken.

According to the BADS manual, each subtest score is standardized into equivalent scores (ranging from 0 to 4). These scores are then summed to derive an index (composite profile score) ranging from 0 to 24. The total profile score can be converted into a standardized score with a mean of 100 and an SD of 15 based on normative data obtained from control participants in the same manner as for IQ scores [55]. This transformation allows separating individuals into six different grades: impaired, borderline, low average, average, high average, and superior [55]. 

#### 2.2.2. Attention and Concentration Battery

The Attention and Concentration Battery [55] consists of a set of computerized tests aimed at measuring different components of attention (Simple Reaction Times; Choice Reaction Times; Auditory, Visual, and Spatial Continuous Performance; Digit Span; Divided Attention (dual task); Color–Word Interference (Stroop test); and Shifting of Attention for Verbal and Visual Targets).

Before starting, instructions are given on how to position the hands on the keyboard. In tests where the participant is only required to press the space bar, he/she is trained to press the space bar by positioning his/her hand or finger. In tests that require the use of both hands, since more than one key must be pressed, specific instructions are given by means of special images that appear on the monitor which show how to position the fingers correctly. This practice can be repeated until the participant has understood the instructions well and has become familiar with the keys to use for the response. If the participant’s condition requires it, the operator can “help” him/her to carry out keyboard operations not directly related to testing (e.g., press the keys to move forward while reading the instructions, or press the key that indicates the end of the test).

The instructions that appear in the first screenshots of each test are formulated for people who have a good command of the language; if there are difficulties in reading or understanding the instructions, they are read by the operator who must make sure—also through the preliminary practice tests—that the participant has understood them. This procedure was adopted whenever appropriate, i.e., in the presence of participants who had reading and comprehension difficulty.

The computer program is structured on three levels of difficulty, differentiated by the number of items contained in each of them and by the duration; the type and functioning of the tests remains the same. Level 1 is meant for 6–9-year-old children, as well as 65–96-year-old adults; Level 2 for 10–18-year-old adolescents; and Level 3 for 19–64-year-old adults. Some second-level tests were used in the present research.

#### 2.2.3. Tests Used

(1)Simple Reaction Times Test

The test consists of pressing the space bar of the computer as quickly as possible when a target (star) appears. Stimuli are presented with randomized times within a 1–3 s range. Median reaction times (RTs) are calculated; the median value is used as it is a measure comparatively insensitive to extreme values.

(2)Choice Reaction Times Test

In order to carry out the test, the numbers from 1 to 6 at the top of the keyboard must be used. Increasing strings of black digits (from 3 to 6 numbers) ordered from the lowest to the highest number appear on the screen and one of the digits becomes red. The task is to press the corresponding key on the keyboard. The target stimulus is presented 60 times, 14 in the 3-digit strings, 16 in the 4-digit strings, 16 in 5-digit strings, and 14 in 6-digit strings. Median RTs for correct responses and percentages of errors and omissions are calculated.

(3)Continuous Performance Test

Three versions of this test are used: auditory, visual, and visuospatial.

*3A*) 
*Auditory Continuous Performance subtest*


In this task the participant has to press the space bar key every time the target letter (O) is spoken. The target letter makes up 30% of the trials (18 out of 60). The presentation is made with randomized times that range from 1 to 3 s.Median RTs for correct responses, percentages of omissions (the key is not pressed when the target letter is heard) and errors (the key is pressed when a letter other than the target is heard) are calculated. 

*3B*) 
*Visual Continuous Performance subtest*


A sequence of randomly occurring symbols is proposed.The task is to press the space bar key every time the target symbol (star) appears on the screen. The target symbol appears in 30% of the trials (18 out of 60). The presentation is made with randomized times that range from 1 to 3 s. Median RTs for correct responses, percentages of omissions (the key is not pressed when the target symbol appears) and errors (the key is pressed to a non-target symbol) are computed. 

*3C*) 
*Spatial Continuous Performance subtest*


An entire page of symbols (36 symbols arranged in 6 lines of 6 symbols) appears on the screen. They are progressively highlighted by a frame that moves with randomized times in a range of 1 to 3 s. On the page there are 12 target stimuli (stars), arranged randomly. The task is to delete the target (star) by pressing the space bar as soon as it is framed. Median RTs for correct responses, percentages of omissions and errors are calculated.

(4)Divided Attention (Dual Task) Test

Here the task is to press a key (space bar) as soon as the target (STAR) appears; at the same time, an entry on the computer reads a list of words and the subject has to press another key (Enter key) every time the target word is spoken (“SOLE”, “SUN”). Thus, attention is distributed over two parallel tasks, i.e., a visual search task and an auditory recognition task. The presentation is carried out with randomized times that range from 1 to 3 s. The list consists of 60 stimuli (images are presented within an oval and whispering words spoken by a computer voice are presented simultaneously). Each of the two target stimuli appear 9 times, i.e., a total of 18 times. The two stimuli never appear at the same time. Median RTs for correct responses, percentages of omissions and errors are calculated.

(5)Color–Word Interference Test (Stroop Test)

This test consists of two sequential tasks. Unlike the other tests, there is no training. The computer must be prepared by applying colored stickers: red on button C, green on button V, blue on button B, black on button N, and brown on button M.

*5A*) 
*Naming colors*


In the first task, which functions as a baseline, participant have to recognize the color of the oval (e.g., press the red button if the oval that appears is red). Sixty stimuli are presented, 12 for each color that appears randomly. The stimuli always appear one second after the previous response is provided and remain on the screen until the new response is given. Median RTs for correct responses and percentages of errors are calculated.

*5B*) 
*Naming word colors*


In the second task, which is the actual interference test, the words that appear indicate a color but are written in a different color (e.g., “red” written in blue); the subjects are instructed to ignore the color indicated by the word and, instead, to press the button corresponding to the color the word is written in (i.e., in the example above, blue and not red).Sixty stimuli are presented, 12 for each color that appears randomly. The stimuli appear one second after the previous response and remain on the screen until the new response is given. Median RTs for correct responses and percentages of errors are calculated.

(6)Shifting of Attention Test

The “shifting” aspect of attention (i.e., the ability to change attention focus when required by the task) is explored by two multiple search subtests that involve the verbal (7A) and the visual–spatial (7B) channel, respectively. 

*6A*) 
*Shifting of Attention with verbal targets*


This task consists of searching and crossing-out3 target letters presented within a display of 30 letters arranged on 5 lines (6 letters per line). Two displays are consecutively presented. The target letters vary from the first to the second display; thus, the focus changes during the task. The letters are written in capitals within a white oval. Proceeding from left to right, the participant must position on the letter by using the right arrow key and cancel it out by using the space bar. At the end of the task, the participant is required to press the Enter key. The total time over the two displays (in seconds) and the percentage of omissions are used as measures of performance.

*6B*) 
*Shifting of Attention with visual targets*


In this test, the targets are small rectangles with a peak of the tail oriented in a different way (perpendicular to the center on the top side, center on the bottom side, center on the left side, center on the right side, diagonal to the top right, the top left, the bottom right or the bottom left). The task consists of searching and crossing-out3 target stimuli on two successive displays, each of which has30 stimuli arranged on 5 lines (6 per line). There are three targets for each display; thus, the participant has to keep them in mind at the same time. Then, they are altered in the second display to force the continuous change of attention focus. Proceeding from left to right, the participant must position on the target stimulus using the right arrow key and cancel it out by using the space bar. It is impossible to go back after the whole quadrant is finished. At the end of the task, the subject has to press the Enter key. The total time over the two displays (in seconds) and the percentage of omissions are used as the performance measure.

### 2.3. Procedure

The tests were administered in two 90-min sessions. In the first session, the WISC-IV or WAIS-IV was administered; in the second session, the BADS and the Attention and Concentration Battery were given. A 15-min break was scheduled during eachof the two sessions.

### 2.4. Data Analysis

Performance on the six tests of the BADS (as well as on its total) was compared in the two groups of participants by using unpaired t tests for a total of 7 comparisons.Based on the Bonferroni correction for multiple comparisons, the alpha *p*-value was adjusted from 0.05 (bi-directional) to 0.0071.The size of the group difference was evaluated by means of Cohen’s *d*, with 0.20, 0.50, and 0.80 as reference points marking small, medium, and large effects, respectively.

Performance on the tests of the Attention and Concentration Battery [55] was compared in the two groups of participants with unpaired t tests. As to RT measures, we checked for the possible presence of RT-accuracy trade-offs, by calculating the correlations among the RT measures and the error and omission values. In the case of errors, only one correlation out of 7 with RTs was significant (Choice Reaction Times: *r* = −0.39, *p* < 0.05) indicating a negative relationship (a direction consistent with an RT-accuracy trade-off). Note that in this experimental condition the proportion of errors was similar in the two groups (see Table 2); therefore, if there was a tendency for RTs to interact with accuracy, this effect was not different in the two groups. In the case of omissions, only one correlation out of 5 was significant (Spatial Continuous Performance test; *r* = −0.47, *p* = 0.01) but positive. Overall, in only one test (Choice Reaction Times) was there some evidence of a possible trade-off between time and accuracy measures. These analyses indicated a limited role of RT-accuracy trade-offs in contributing to group differences in RT measures. Considering different dependent measures (RTs in ms, times in s, and percentages of errors and omissions) a total of 24 comparisons were carried out. Based on the Bonferroni correction for multiple comparisons, the alpha *p*-value was adjusted to 0.002. Effect size was evaluated by means of Cohen’s *d*.

We also evaluated the presence of global components in the attention data limited to the conditions with RTs. The assumption is that, in RT data, there is a co-variation of SD with means such that more difficult condition means are systematically associated with greater individual differences [47]. This can be evaluated by carrying out a plot contrasting means against SDs. Based on previous data [47], a slope of ca. 0.30 is expected. According to Myerson et al. [47], the presence of this relationship is crucial for detecting the presence of global components in the data. Note also that this relationship constitutes a systematic variation from the standard assumption of homogeneity of variance (used as a basis for parametric comparisons). Furthermore, according to Myerson et al. [48], the intercept of the regression over the *x*-axis represents an estimate of the sensory–motor component of the task (in several experiments a value near 300 ms is reported).

It is also possible to examine whether a global factor accounts for all (or part) of the group differences across the RT conditions considered. For this purpose, a Brinley plot contrasting the mean performances of the two groups on the same conditions can be used. The presence of a global factor is marked by a single regression line accounting for a sizeable proportion of the variance in the data. The slope of the regression is an indication of the strength of the global component contributing the group differences over and above the effect of specific experimental manipulations. It is also expected [47] that the regression will have a negative intercept and that the “crossing” of the regression line over the diagonal line marking iso-performance between the two target groups represents an estimate of the sensory–motor component of the task, hypothesized to be similar to the one detected in the plot contrasting means against SDs.

## 3. Results

### 3.1. Intelligence Tests

On the WISC-IV, the participants with ID had an average IQ level of 56.83 (DS = 7.9); on the WAIS-IV, the average IQ level was 60.46 (DS = 6.92). The control group presented an average IQ level of 101.83 (SD = 17.52) on the WISC-IV, and an average IQ level of 107.46 (SD = 13.47) on the WAIS-IV.

Regarding the index scores, on the WISC-IV, we found the following: Verbal Comprehension Index (VCI), M = 64.66, DS = 10.49; Perceptual Reasoning Index (PRI), M = 68.25, DS = 9.21; Working Memory Index (WMI), M = 64.66, DS = 10.49; Processing Speed Index (PSI), M = 74.91, DS = 13.38. On the WAIS-IV, we found the following: VCI, M = 67.66, DS = 17.02; PRI, M = 72.6, DS = 10.22; WMI, M = 67.33, DS = 6.45; VCI, M = 71.73, DS = 12.09. 

### 3.2. Executive Tests

Mean performances on the BADS tests for the group of individuals with ID and the control group are presented in Table 1. Note that individuals with ID performed worse than controls on all subtests and thus also obtained a lower total score. Inspection of *d* values indicates large effect sizes in all cases. The largest *d* value (1.52) was present in the case of the Rule Shift Card test and the smallest (0.87) in the case of the Temporal Judgement test. The *d* value for the total score was very high (2.34), i.e., it was higher than any of the single subtest values.

For the sake of presentation, the total score is also presented in terms of standardized values. In general, adolescents and young adults with ID showed a severe impairment with 10 individuals (out of 27) scoring 65 or less. However, considerable individual differences were also present: Based on the published cutoffs, five individuals were in the defective range (60 or below), five in the borderline range (65), nine in the low-normal range (70–80), and eight in the normal range (85–115), with five of them actually scoring above 100. We examined whether there is a difference in IQ between the subjects who scored in the normal range (*N* = 8), as compared to the rest of the group with ID (*N* = 19).The average IQ of participants scoring in the normal range in the BADS was 61.8 (SD = 5.7), while that of the individuals who achieved lower scores was 57.3 (SD = 7.9); the difference was not statistically significant (*t* = 1.33, n.s.).

### 3.3. Attention Tests

Performance on the various attention tests of the Attention and Concentration Battery [55] is presented in Table 2. Results for different measures (RTs in ms, times in s, errors, and omissions) are presented separately.

When Student’s *t*-test values were corrected for multiple comparisons, the two groups were different on a limited set of measures/tests: Adolescents and young adults with ID had slower RTs in the Choice Reaction Time test, as well as in the Color–Word Interference Test (both Color Naming and Color Interference conditions). They were also slower in the Shifting of Attention test (both in the condition with verbal targets and in that with visual targets). By contrast, the two groups were very similar (and not significantly different) in the Simple Reaction Times test, the Continuous Performance (all three conditions), and in the Divided Attention test. In the case in which groups were significantly different, the effect size was always large (with *d* values in all cases >1).

None of the error measures (whether errors or omissions) yielded significant group differences, although some group differences would have been significant with standard *p* values <0.05.

Limited to tests with RT measures, the performance of the two groups of participants is also illustrated in Figure 1. The figure clearly shows that the group difference is present only for some of the attention conditions.

### 3.4. Analysis of Global Components in RT Tasks

To evaluate global components in the RT attention data the plot contrasting means against SDs in the same conditions is presented in Figure 2A. Both values related to the ID sample and the control sample are considered in the linear regression. Inspection of the figure indicates that individual variability (i.e., SD) increases as a function of condition difficulty with a 0.23 slope. This relationship accounts for 0.69 of the variance. The intercept on the *x*-axis indicates 142 ms as an estimate of the sensory–motor component of the response.

The Brinley plot contrasting the mean performances of the two groups on the same conditions is presented in Figure 2B. In this graph, the diagonal indicates equal performance of the two groups. Values above the diagonal indicate worse performance of the group of individuals with ID. The linear regression indicates a moderate slope (1.32). The variance accounted for by the linear regression is 0.89. Note that the regression line crosses the diagonal line at about 390 ms, which also represents an estimate of the sensory–motor component of the response according to [47]. 

## 4. Discussion

With regard to EF, adolescents and young adults with ID were severely impaired across all EF tests administered. Effect sizes indicated somewhat greater effects in the case of tasks calling for cognitive flexibility (Rule Shift Card subtest), problem-solving (Action Programme subtest), and navigational planning (Zoo Map Test subtest), but all EF processes were clearly affected. Furthermore, the total score of the BADS battery yielded the largest effect size in the comparison between the two groups. It is well-known that different EF processes are only moderately related to each other [4]. Therefore, it appears that the total score of the battery captures the group difference best, as it includes a variety of different EF processes, each partially contributing to the overall group effect. 

It can be added that an ecological battery such as the BADS seemed to capture well the performance of individuals with mild ID on EF tasks. Previous research has yielded mixed results on this point. Chevalère and colleagues [13] used the BADS battery successfully inchildren with Prader–Willi syndrome and ID, whereas Willner et al. [57] reported floor performance using the children’s version of the scale (BADS-C). The IQ ranges of these studies are similar, but the age ranges differ appreciably as the participants in Willner et al.’s study [57] were considerably younger (Mean = 10.8 years) than those in Chevalère et al.’s [15] study (Mean = 28.05 years) and in the present study (M = 17.4).Therefore, it appears that the BADS test is appropriate to use with individuals with ID, but only from a given age; note also that the normalization of the battery considers 16 years of age as a lower limit for administration [56]. 

Overall, these results indicate that individuals with ID present an across the board deficit in EF which is best captured by a measure that sums performance on a variety of EF measures. The presence of considerable individual differences was also detected. Thus, approximately 30% of the participants with ID (8 out of 27) scored well within the normal limits (i.e., above 85) and some of them also above 100.

By contrast, regarding the attention test, adolescents and young adults with ID did not show an across the board deficit in attention processing. In several tests (Simple Reaction Times test, the Continuous Performance test (all three conditions), and the Divided Attention test) performance of the two groups was remarkably similar. These results indicate that individuals with ID are not generally slower but present specific deficits in some attention conditions. In particular, individuals with ID were not slower in terms of RTs to simple visual stimuli and in the case of a test calling for continuous performance. These findings indicate no deficit regarding the “intensive” dimension of attention. By contrast, individuals with ID are impaired in conditions which call for interference (such as the Color–Word Interference Test, or Stroop test) or the shifting of attention (Shifting of Attention for Verbal and Visual Targets). These findings indicate specific deficits in tasks mapping the selectivity dimensions of attention.

However, some caveats to this conclusion must be noted. First, individuals with ID did not show a deficit in the Divided Attention test, which would be expected if the selectivity dimension marks the deficit. In general, both groups of participants performed this task well and with very few errors (and omissions). It is possible that this task is not as demanding in terms of attentional resources and that experimental conditions with a tighter time schedule between stimuli would yield different results. Further research is needed to examine this possibility. Second, individuals with ID were impaired in both conditions of the Color–Word Interference Test (Stroop test), i.e., both in the case in which naming the color was interrupted by the automatic activation of the name code and in the simpler case in which the color of the visual stimulus had to be named but no interference was present. In keeping with the selectivity hypothesis, the group difference in RTs was quantitatively greater in the former case. Still, there is no obvious explanation of the clear group difference in the baseline condition. It should be noted that Danielsson et al. [58] reported the presence of verbal fluency problems and interpreted this as indicating a deficit in speed of accessing lexical items and difficulty with working memory-related executive control at encoding. Thus, it is possible that the slower performance in naming colors indicates an additional and different deficit from the difficulty in inhibiting prepotent responses, a key marker of EF [4].

Analysis of the global components in tasks with RT measures generally confirmed the specificity of the attention deficits of individuals with ID. First, results indicated that, as expected, there was a moderate covariation between means and SDs with more difficult tasks yielding greater individual variability over and above the specific characteristics of the tasks and of group belonging. The value of the slope of the relationship (0.23) is similar, though slightly smaller, than that typically reported for this type of task (ca. 0.30; Reference [47]). These data are in keeping with the idea that, as expected, global components are present in the data.

Analysis with the Brinley plot indicated generally larger group differences on more difficult tasks; however, the slope factor (1.32) was rather small. In considering the size of this factor, one should also consider that based on models such as the DEM [47] this multiplicative factor applies to only a part of the response time. Thus, Myerson et al. [47] proposed that the overall response consists of two parts, one of which is a sensory–motor component which contributes to the response as a constant value. In the particular case of the present data, this sensory–motor component was estimated as 142 and 390 ms in the two plots considered (Note that these two values are expected to be consistent based on model predictions [47]). In their studies on aging, Myerson et al. [47] reported a value of about 300 ms across a large variety of tasks. Therefore, the present data are broadly in the expected range, though some instability was present between the two estimates, possibly because of the limited number of data points considered. The residual part of the response represents the decisional (or central) compartment and is sensitive to the multiplicative factor identified as the slope value in the Brinley plot. Therefore, based on these considerations, it seems that individuals with ID are impaired on a 1.32 factor limited to the decisional component of the response, i.e., a relatively small deficit compared to the large impairment in EF. Furthermore, inspection of the Brinley plot did not indicate a clear gradual increase in group differences as a function of condition difficulty. Rather, the two groups appear virtually identical on easier tasks and show group differences on some more difficult tasks. Therefore, these data do not seem to be in line with the idea that a global factor (i.e., cognitive speed) contributes to the group differences over and above the specific effects due to the experimental manipulations. By contrast, they seem more in line with the conclusion that, whenever present, group differences are selectively associated with specific task manipulations, not to global differences in cognitive speed.

This conclusion is at variance with that put forward by Kail [49] who carried out a large meta-analysis of studies comparing the performance of individuals with intellectual disability and control participants and concluded that “these results are consistent with the view that differences in processing speed between persons with and without mental retardation reflect some general (i.e., nontask specific) component of cognitive processing”. Comparing the present results to the analyses carried out by Kail [49] is not straight forward. On one hand, Kail [49] considered a number of studies carried out in the sixties and seventies and we have seen that research in that era has been severely criticized due to various, important methodological pitfalls [39]. Thus, to the extent to which studies included controls who were not matched for significant variables (such as institutionalization), it is conceivable that the overall estimates reported by Kail [49] may be somewhat amplified. On the other hand, a close inspection of values indicates that, at least for some age ranges, the two sets of data are not so far apart. In the present sample of 15.1-to-23-year-old adolescents and young adults, the slope factor was 1.32. Kail [49] reported a similar value of 1.43 for 12-to-16-year-old adolescents; however, appreciably higher values were reported for the 17–33 age range. In this latter case, he only reported separate values for groups with different IQ ranges, e.g., 2.34 for young adults with an IQ between 50 and 63.5, and 1.76 for individuals in the 64–67 IQ range.

It is difficult to reconcile these two sets of analysis because not only tasks but also primarily participant selection procedures were very different. We propose the following tentative considerations: First, the present results raise the possibility that the impairment in global processing was overestimated in previous research and, second, that the presence of global slowing might be present but small in comparison with the clear deficit shown by individuals with ID on attention tasks calling for a selective component. Certainly, further research is warranted before these two proposals can be accepted. In particular, it should be noted that global components are best estimated by using large sets of data. Therefore, this aspect warrants further examination.

Finally, we believe that the present findings could have interesting implications for the evaluation of individuals with ID, in particular in view of the important consequences of EF and attentional deficiencies on everyday life activities. In general, it appears that data on both domains emphasize the importance of an accurate diagnostic process. In this vein, it is of note that a significant proportion of adolescents and young adults showed a spared performance on EF tasks. Furthermore, attention deficits were also quite selective. Therefore, it is important that EF and attentional testing be accurately carried out to guide interventions and management of individuals with ID.

## 5. Conclusions

Overall, the present research confirms the presence of large group differences in executive function between adolescents and young adults with ID and chronologically matched controls. All major areas of EF appear to contribute to this relationship, even though considerable individual variability is present and several adolescents and young adults with ID performed well within normal limits. Compared to EF impairments, the deficits in attentional processing were considerably more selective and pertained mostly to tasks which taxed the selective dimension of attention with performance on intensive tasks being almost entirely spared. Moreover, data on individual speed of processing (independent of task) indicated very moderate slowing. In any case, this finding must be confirmed in future research, as it differs appreciably from the results of the only meta-analysis available on the topic [50]. We propose that the examination of EF and attention process in the same group of participants is a strength of the present study, in that the differential findings between EF and attention processes cannot be easily attributed to sampling biases. 

## Figures and Tables

**Figure 1 brainsci-11-00042-f001:**
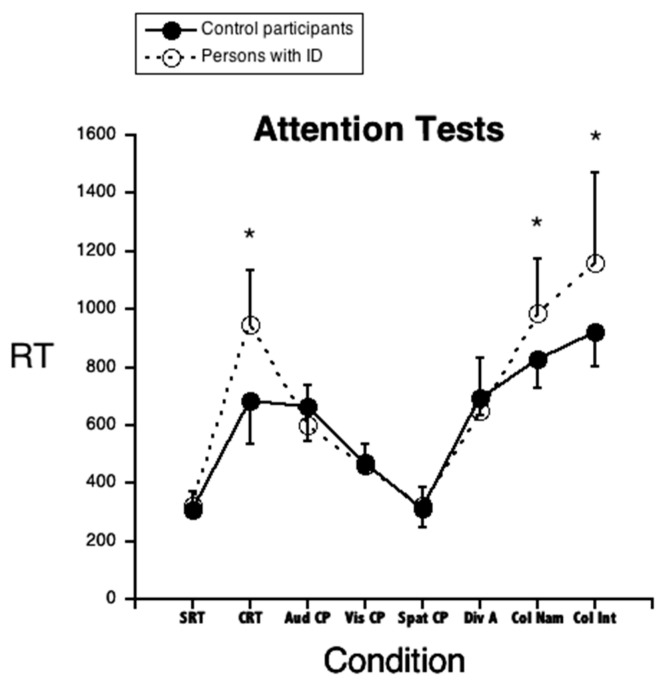
Mean performance (and SDs) of individuals with ID and controls on the various tests of Table 2 Legend: SRT, Simple Reaction Times; CRT, Choice Reaction Times; Aud CF, Auditory Continuous Performance; Vis CP, Visual Continuous Performance; Spat CP, Spatial Continuous Performance; Div A, Divided Attention; Col Nam, Color Naming; Col Int, Color–Word Interference. The asterisk (*) mark group differences which are statistically significant.

**Figure 2 brainsci-11-00042-f002:**
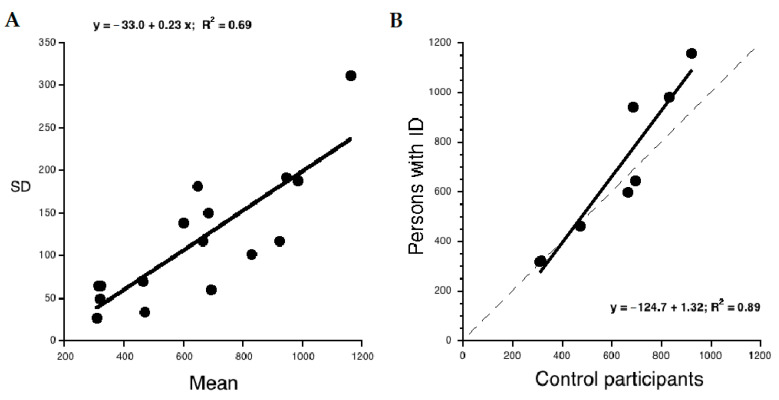
(**A**) Standard deviations (SD) across individuals for each group and the experimental conditions of the Attention and Concentration Battery [54] that yield RT measures are plotted as a function of the corresponding means. The equation of the linear regression is also reported. (**B**) Brinley plot reporting mean RTs for individuals with ID and controls in various conditions. The dashed line indicates equal performance between the two groups. The equation of the linear regression is also reported.

**Table 1 brainsci-11-00042-t001:** Mean performance (and SD) in the Behavioral Assessment of Dysexecutive Syndrome (BADS) tests of individuals with intellectual disability (ID) and controls. The *p*-values associated with Student’s *t*-tests are presented; based on the Bonferroni correction for multiple comparisons, *p* < 0.007 are considered significant. Effect sizes (Cohen’s *d* values) are also reported.

	Individuals with ID		Controls			
Condition	Mean	SD	Mean	SD	*p*	*d*
Rule Shift Card	2.04	1.16	3.44	0.70	0.00000	1.52
Action Programme	3.00	0.96	3.89	0.32	0.00007	1.39
Key Search	0.96	0.94	2.00	1.00	0.00026	1.07
Temporal Judgement	2.11	1.12	3.00	0.92	0.00249	0.87
Zoo Map Test	1.89	0.97	3.11	0.89	0.00001	1.31
Modified Six Elements	2.30	1.46	3.44	0.75	0.00082	1.04
Total	12.30	3.68	18.85	1.92	0.00000	2.34
IQ score	76.48	18.39	109.26	9.58	0.00000	2.34

**Table 2 brainsci-11-00042-t002:** Mean performance (and SD) in the Attention and Concentration Battery [54] of individuals with ID and controls. Reaction times (RTs) (in milliseconds (ms)), times (in seconds (s)), and percentages of errors and omissions are presented separately.The *p*-values associated with Scheme 0 are considered significant and are marked with an asterisk (*). Effect sizes (Cohen’s *d* values) are also reported.

	Individuals with ID		Controls			
Test/Measure						
RTs (ms)	Mean	SD	Mean	SD	p	d
Simple Reaction Times	320	50	308	27	0.28	0.31
Choice Reaction Times	944	193	684	151	0.000001 *	1.51
Auditory Continuous Performance	601	139	664	118	0.08	−0.49
Visual Continuous Performance	463	70	469	34	0.69	−0.11
Spatial Continuous Performance	323	65	313	65	0.57	0.15
Divided Attention	648	182	693	60	0.23	−0.37
Color Naming	984	189	829	102	0.0006 *	1.06
Color–Word Interference	1161	312	921	118	0.0007 *	1.12
**Time measures (s)**						
Shifting of Attention (verbal targets)	61.7	12.5	46.4	13.7	0.00008 *	1.17
Shifting of Attention (visual targets)	88.3	21.4	66.4	11.4	0.00003 *	1.33
**Errors**						
Choice Reaction Times	3.19	3.48	2.30	1.90	0.25	0.33
Auditory Continuous Performance	0.70	0.99	0.15	0.36	0.01	0.82
Visual Continuous Performance	0.78	1.37	0.26	0.53	0.07	0.55
Spatial Continuous Performance	0.26	0.59	0.15	0.60	0.50	0.19
Divided Attention	1.15	1.41	0.56	0.85	0.07	0.53
Color Naming	2.19	2.13	1.00	0.88	0.01	0.79
Color–Word Interference	3.30	3.22	1.96	3.47	0.15	0.40
**Omissions**						
Choice Reaction Times	1.93	2.79	0.41	0.80	0.01	0.85
Auditory Continuous Performance	0.11	0.32	0.00	0.00	0.08	0.69
Visual Continuous Performance	0.26	0.53	0.11	0.32	0.22	0.35
Spatial Continuous Performance	0.07	0.27	0.11	0.42	0.70	−0.11
Divided Attention	1.33	1.33	0.48	0.85	0.007	0.78
Shifting of Attention (verbal targets)	3.19	2.43	1.70	2.03	0.02	0.66
Shifting of Attention (visual targets)	3.11	2.74	1.63	1.55	0.02	0.69

## Data Availability

The data presented in this study are available on request from the corresponding author. The data are not publicly available as not data repository is yet available in our institute.

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
