# Peer review of "Executive Functions and Attention Processes in Adolescents and Young Adults with Intellectual Disability"

_brainsci, 2021, doi:10.3390/brainsci11010042_

Round 1

Reviewer 1 Report

Review comments

This study investigates executive functions and attention processes in individuals with intellectual disabilities (ID), by applying the Behavioural Assessment of Dysexecutive Syndrome (BADS) battery, and a range of attention tests to a group of persons with ID and an age matched control group. The results of the study shows that the subjects with ID performs worse than the age-matched controls without ID, BADS-battery indicating impaired executive functions. The ID-group only showed reduced attention on some measures, in comparison with the control group.

General comments

The study includes a significant amount of work. One major consideration is to use the IQ-data. It would be of interest to analyze if there is any correlation between the subjects’ IQ and their performance in BADS and the attention tests. E.g., does the subjects in the ID-group that score in the normal range on the attention tests also present higher IQ scores within the group?

The authors nicely describes the testbattery but it doesn't clear what the timescale for the substantial testing. Was it performed over several days and if so, then it would be nice to have a better understanding of that via a description or an illustration.

Below I have more specific comments for the text.

Specific comments:

Abstract:

Line 19-23: Results are described without any actual test results. It would be beneficial to the reader if the authors included test scores for the two groups on the tests that are mentioned in the abstract.

Introduction:

Line 29: Reference(s) to the definition of executive functions would be preferable. Otherwise a statement that underlines that what is written is the authors own definition.

Line 37-38: Statements without any reference. In addition, what form of adaptation is executive functions fundamental for in the authors opinion? Behavioural? Motor? Environmental?

Line 46: FE? Must be a typing error.

Line 54.58: The claims made in this sentence are not supported by the reference. The study by Griffth et el. (1999), investigates aspects of executive functions in children with autism spectrum disorders, and does not study the terms mentioned (e.g. self-control, emotional instability, careless behavoir, impulsiveness, hygiene)

Line 59-61: Language is a bit off. If the area have been the object of much research, there must be more than one reference. Is returning to ones work life after a traumatic brain injury considered an everyday activity?

Line 62-64: I would like a reference on that statement put forward by the authors in this sentence.

Line 65-68: This sentence is plagiarized from the introduction in the referenced article: Japundža-Milisavljevic, M.; Macešic-Petrovic, D. Executive functions in children with intellectual disabilities. Brit. J. Dev. Dis. 2008, 54(107), 113-121

Line 77: typing error on EF.

Line 78:”Fragile X syndrome”, is usually the way it is denoted in English writing.

Line 92-94: What kind of performance do the authors refer to here? Please, elaborate a bit and add references.

Line 121: Rephrase the sentence, especially “can be seen as due”. ‘

Line: 150-152: You mention earlier studies and their work, but the reference is to a textbook. Could you provide references for the studies you mention?

Line 152-154: Please cite the review mentioned in this sentence.

Line 162: “RF capacities”, must be a typing error

Materials and Methods:

Line 169-187: The results of the intelligence tests should be presented in the results-section. In addition, it would be preferable to write out the abbreviations of the subtests of the WISC and WAIS tests.

Line 243: The software used should be explicitly referenced in the text. This Applies to the company who have published the BADS test as well.

Line 251: What does this “help” consist of more specifically?

Line 256: The sentence “This procedure was adopted whenever appropriate” is a bit unclear. Please, elaborate on the procedure for making sure that the test procedures were understood by the test person. This is an important ethical consideration.

Line 268: How are the subjects instructed to keep their hands between trials?

Line 329-332: This section could use some clarification: What do you mean by “cancelling” the letters? Is this how the subjects responds to the target letters? You mention lines, and quadrants. Is six lines of five letters what you mean by a “quadrant”? What does the enter key do exactly?

Results:

Line 385-389: Some repletion in this paragraph. The last part alone is sufficient.

Line 415: Please indicate in figure 1 the values that are significant with asterisks (*).

Line 435: Figure text. It would be good to elaborate on how the numbers in figure 2a is generated. Is it the mean of both groups –i.e. pooled data?

Discussion:

Line 446: Is it the different aspects of executive functions, you refer to here? This in unclear. Please, refrase and elaborate the sentence.

Line 458: Reference 55 is the normalization study.

Line:512: FE?

Line 520: The language could use a revision here.

Author Response

Please, see the attachment with our responses to Reviewer 1

Reviewer 2 Report

This is an interesting study that examines executive function (EF) deficits in individuals with intellectual disability. According to the results, EF in the specific population are not globally affected, since performance tapping into the intensive dimension of attention tasks was almost entirely spared.

I only have few concerns with respect to the explanation of the intensive and selective dimension of attention in the Introduction and the reaction time measurements in data analysis. More specifically:

lines 40-44:it is important that the authors elaborate upon and sufficiently explain what the intensive and selective dimension of the attention processes is. Also, provide examples.

Materials and Methods

1) Which were the exclusion criteria for selecting the control individuals? Were there any specific tests used (e.g. interviews, medical history questionnaires, language tests, such as picture naming)? For example, how were the authors sure that none of the controls had a developmental language disorder, high functioning autism, or in any case, a disorder that could have affected their performance in the EF tasks of the study?

2) Regarding the RT of the attention tests, were there any outliers? I assume that there were outliers since the sample of the participants was large. How have the authors trimmed RTs to remove outliers? Outliers should have been removed before data analysis.

3) Did the authors test for any RT-accuracy trade-off effects in the tests measuring both accuracy and reaction times? RT-accuracy trade-off effects should be tested before analyzing the data.

Minor points

Abstract

line 19. please provide the raw number of the individuals corresponding to the 30% percentage, i.e. ?? individuals out of ?? individuals in total

line 20. Specify the attention tasks that were deficient.

Be consistent with the use of the abbreviation EF throughout the manuscript; sometimes it is used as FE

lines 82-85: greater deficits in comparison to which population? Please, specify.

line 100: the word "is" is repeated twice

lines 178-182: Provide the full names of all the abbreviations of the tests used

Author Response

Please, see the attachment with our responses to Reviewer 2

Round 2

Reviewer 2 Report

I am very satisfied with the authors' responses to my concerns.